Disentangling bias for non-destructive insect metabarcoding

http://orcid.org/0000-0001-8064-4460 Martoni Francesco 1 francesco.martoni@ecodev.vic.gov.au
http://orcid.org/0000-0002-0664-7564 Piper Alexander M. 1 2
Rodoni Brendan C. 1 2
http://orcid.org/0000-0001-7864-5712 Blacket Mark J. 1
1 Agriculture Victoria Research, AgriBio Centre for AgriBioscience, State Government Victoria , Bundoora, Victoria , Australia
2 School of Applied Systems Biology, La Trobe University , Bundoora, Victoria , Australia
Colla Sheila
Electronic publication date: 2022 Feb 23
Publication date: 2022
Volume: 10
Electronic Location ID: e12981
Received 2021 Oct 1; Accepted 2022 Feb 1
Copyright: © 2022 Martoni et al.
Copyright year: 2022
Copyright holder: Martoni et al.
License: This is an open access article distributed under the terms of the Creative Commons Attribution License, which permits unrestricted use, distribution, reproduction and adaptation in any medium and for any purpose provided that it is properly attributed. For attribution, the original author(s), title, publication source (PeerJ) and either DOI or URL of the article must be cited.
License URL: https://creativecommons.org/licenses/by/4.0/

Keywords: High throughput sequencing, Barcoding, Biodiversity, Primer bias, Entomology, Biosecurity

Funding: iMapPESTS project Horticulture Innovation Australia ST16010 Australian Government Department of Agriculture as part of its Rural R & D for Profit program Grain Research Development Corporation (GRDC) This work was supported by the iMapPESTS project, supported by Horticulture Innovation Australia (ST16010) through funding from the Australian Government Department of Agriculture as part of its Rural R & D for Profit program and Grains Research and Development Corporation. The funders had no role in study design, data collection and analysis, decision to publish, or preparation of the manuscript.

==============================
A fast and reliable method for obtaining a species-level identification is a fundamental requirement for a wide range of activities, from plant protection and invasive species management to biodiversity assessments and ecological studies. For insects, novel molecular techniques such as DNA metabarcoding have emerged as a rapid alternative to traditional morphological identification, reducing the dependence on limited taxonomic experts. Until recently, molecular techniques have required a destructive DNA extraction, precluding the possibility of preserving voucher specimens for future studies, or species descriptions. Here we paired insect metabarcoding with two recent non-destructive DNA extraction protocols, to obtain a rapid and high-throughput taxonomic identification of diverse insect taxa while retaining a physical voucher specimen. The aim of this work was to explore how non-destructive extraction protocols impact the semi-quantitative nature of metabarcoding, which alongside species presence/absence also provides a quantitative, but biased, representation of their relative abundances. By using a series of mock communities representing each stage of a typical metabarcoding workflow we were able to determine how different morphological (i.e., insect biomass and exoskeleton hardness) and molecular traits (i.e., primer mismatch and amplicon GC%), interact with different protocol steps to introduce quantitative bias into non-destructive metabarcoding results. We discuss the relevance of taxonomic bias to metabarcoding identification of insects and potential approaches to account for it.

Introduction

Species identification is a fundamental pre-requisite for basic and applied ecology. In the field of entomology, species-level identification is required for biodiversity assessments and checklists (Girón & Short, 2021), understanding ecology and behavior (Lefort et al., 2020), forensic investigation (Pohjoismäki et al., 2010), taxonomy (Schutze et al., 2017), and management of agricultural pests. Invasive insect species are becoming a major threat to agroecosystems (Paini et al., 2016), with biological invasions becoming one of the main menaces to agricultural production (Meyerson & Mooney, 2007; Hulme, 2009; Chown et al., 2014). Therefore, extensive trapping and monitoring activities to detect new insect invasions are carried out both on agricultural properties (Low-Choy, 2015) and protected environments, such as National Parks (e.g., Davidovitch et al., 2009). This surveillance is leading to an increasing demand for species-level identification of large volumes of insects trapped for plant protection, biosecurity and agriculture (Piper et al., 2019).

However, availability of taxonomic expertise for insect identification is extremely limited, with often only a few experts worldwide per taxonomic group. Therefore, a range of molecular techniques have been developed to allow more standardized identification of insect species by non-specialists (Piper et al., 2019). Most notably, the DNA barcoding technique (Hebert et al., 2003) allows comparison of a short standardized genetic region from an unidentified specimen to a vast number of known species deposited in reference databases. Generally, invertebrate barcoding studies have targeted the subunit 1 of the mitochondrial cytochrome c oxidase gene (COI) for specimen identification (e.g., Andújar et al., 2018; Elbrecht, Peinert & Leese, 2017; Yu et al., 2012). DNA barcoding is regularly applied to identification of undescribed species native to localised areas (Martoni, Taylor & Blacket, 2020), as well as to identification of invasive insect pests (Armstrong & Ball, 2005), and is now widely accepted within plant pest diagnostics protocols (EPPO, 2021; Ashfaq, Hebert & Naaum, 2016). However, difficulties remain scaling this approach to the sheer volume of specimens that can be caught in a surveillance trap, or in a field sampling season. Therefore, with the advent of high throughput sequencing technologies, the focus is now shifting from the single specimen sequencing of DNA barcoding to identifying entire communities of species in parallel using DNA metabarcoding (Piper et al., 2019 and references therein).

The semi-quantitative nature of metabarcoding data has led to concerns around the appropriateness of this technology for surveillance, biomonitoring and assessment of pest pressures (Darling et al., 2020; Martins et al., 2019). Metabarcoding assays only provide relative abundance data, with sequence reads returned for a species only meaningful relative to the rest of the taxa within the sample (Gloor et al., 2017). In addition to quantification issues relating to sample composition, taxonomic biases can be introduced during laboratory processing of samples, including during DNA extraction procedures and genetic marker isolation and amplification. These biases arise due to species-specific differences in morphological and molecular traits which interact with steps of the laboratory protocol to preferentially detect certain taxa at the expense of others. DNA extraction from complex insect communities for metabarcoding analysis has often involved destructive homogenisation of tissues, which results in larger-sized organisms contributing a larger quantity of DNA molecules to the DNA extraction pool than smaller-bodied insects (Elbrecht, Peinert & Leese, 2017). Nevertheless, when individual specimen size is accounted for through size sorting prior to DNA extraction, the influence of primer-template mismatch generally outweighs DNA extraction bias for macroinvertebrates (Braukmann et al., 2019; Elbrecht & Leese, 2015), at least for destructive homogenisation-based DNA extraction.

Recently, non-destructive DNA extraction has emerged as an alternative to homogenisation-based methods in order to retain voucher specimens for morphological confirmation of metabarcoding detections (Carew, Coleman & Hoffmann, 2018; Martins et al., 2019; Batovska et al., 2021). This is of particular importance in the context of agricultural biosecurity and other regulatory applications of metabarcoding, allowing DNA sequences to be linked to an insect sample, which can be preserved in entomological collections for future records (Martoni, Valenzuela & Blacket, 2021). While it is well established for homogenised samples that large organisms will contribute a higher abundance of DNA than small organisms, this may differ for non-destructive metabarcoding, where contribution of DNA may instead depend on surface/volume ratio of insect to extraction buffer (Marquina et al., 2019). Furthermore, differences in sclerotization of exoskeletons could affect permeability of DNA and impact detection efficiency (Carew, Coleman & Hoffmann, 2018; Marquina et al., 2019). Moreover, with this move from destructive to non-destructive DNA extraction it is unclear if earlier results and assumptions about the bias-generating process still hold.

Here we compared two different non-destructive DNA extraction methods previously tested for their capability to extract DNA from preserved trapped insects (Martoni et al., 2021). We applied these to mock communities composed of a mix of possible insect pests and harmless by-catch species, following a typical metabarcoding protocol from the recent literature, and using two combinations of degenerate PCR primers. We aimed to measure the taxonomic bias introduced by the DNA extraction, PCR, and library preparation stages, and evaluate the downstream effects on the two main diagnostic-related aspects: sensitivity and quantitation.

Materials and Methods

Samples and morphological traits

For this study we used adult specimens belonging to 16 insect species (Table 1). Of these, 15 species were obtained from insect colonies reared at the AgriBio laboratory of Agriculture Victoria, while another species (Acizzia sp.) was field-collected (Table 1). Insect specimens from the colonies were preserved in absolute ethanol and deposited into the Victorian Agricultural Insect Collection (VAIC). Measurements to obtain the volume size were taken from 10 individuals for each species and the result was averaged in Table 2. Measurements were taken using the Leica Application Suite software v4.5.0, from five to 20 images were stacked to generate each photo using a Leica stereo microscope M205C with a DFC450 camera. For each insect, images were taken from dorsal, lateral, and frontal view in order to obtain a measurement of the volume (length * width * depth) of head, thorax and abdomen. Hardness of the exoskeleton was estimated by attempting to compress and pierce it using forceps, and the 16 species were assigned to three discrete categories (soft, intermediate, hard) (Table 2). Specimens from all taxa were then grouped into eight insect pools, with six pools containing 100 individual insects and the same 15 species, while another two pools also contained a single specimen of the Bradysia nr. ocellaris, for a total of 16 species, and 101 individuals (Table 1).

Table 1 Composition of the eight pools used for this study.

		Pool	
Species	Order	1	2	3	4	5	6	7	8	
Carpophilus davidsoni	Coleoptera	25	50	5	10	25	50	5	10	
C. truncatus	Coleoptera	1	1	1	1	1	1	1	1	
Bactrocera tryoni	Diptera	1	1	1	1	1	1	1	1	
Bradysia nr. ocellaris	Diptera	0	0	0	0	1	0	1	0	
Drosophila hydei	Diptera	1	1	1	1	1	1	1	1	
D. melanogaster	Diptera	5	10	25	50	5	10	25	50	
D. simulans	Diptera	1	1	1	1	1	1	1	1	
Scaptodrosophila lativittata	Diptera	1	1	1	1	1	1	1	1	
Acizzia alternata	Hemiptera	1	1	1	1	1	1	1	1	
A. solanicola	Hemiptera	10	25	50	5	10	25	50	5	
Acizzia sp.	Hemiptera	1	1	1	1	1	1	1	1	
Diuraphis noxia	Hemiptera	1	1	1	1	1	1	1	1	
Metopolophium dirhodum	Hemiptera	1	1	1	1	1	1	1	1	
Rhopalosiphum padi	Hemiptera	1	1	1	1	1	1	1	1	
Aphidius colemani	Hymenoptera	36	5	6	12	36	4	9	20	
Lysiphlebus testaceipes	Hymenoptera	14	0	4	13	14	1	1	5	
Total individuals	100	100	100	100	101	100	101	100	
Note:

The number of individual insects is reported for each pool, as well as the total number of individuals (in bold). DNA and PCR pools were assembled with the same proportions reported here.

Table 2 Molecular and morphological characteristics of the insects used in the pools.

		Molecular traits	Morphological traits	
Species	Order	fwhF2 mismatch	fwhR2n mismatch	Amplicon GC%	fwhF2 mismatch	HexCOIR4 mismatch	Amplicon GC%	Volume (mm3)	Hardness	
Carpophilus davidsoni	Coleoptera	0.12	0.09	0.37	0.12	0.06	0.38	3.80	3	
Carpophilus truncatus	Coleoptera	0.04	0.04	0.35	0.04	0	0.35	3.91	3	
Bactrocera tryoni	Diptera	0	0.04	0.37	0	0	0.37	26.51	2	
Bradysia nr. ocellaris	Diptera	0.08	0	0.34	0.08	0	0.34	0.25	2	
Drosophila hydei	Diptera	0.04	0	0.33	0.04	0.06	0.33	2.19	2	
Drosophila melanogaster	Diptera	0	0	0.30	0	0	0.30	1.14	2	
Drosophila simulans	Diptera	0	0	0.32	0	0	0.32	0.79	2	
Scaptodrosophila lativittata	Diptera	0	0	0.28	0	0	0.28	3.09	2	
Acizzia alternata	Hemiptera	0	0.04	0.30	0	0	0.31	0.22	1	
Acizzia solanicola	Hemiptera	0	0.04	0.32	0	0	0.33	0.47	1	
Acizzia sp.	Hemiptera	0.08	0.09	0.32	0.08	0	0.33	1.18	1	
Diuraphis noxia	Hemiptera	0.04	0	0.20	0.04	0	0.20	0.50	1	
Metopolophium dirhodum	Hemiptera	0.04	0	0.22	0.04	0	0.22	0.65	1	
Rhopalosiphum padi	Hemiptera	0.04	0	0.19	0.04	0	0.20	0.18	1	
Aphidius colemani	Hymenoptera	0	0	0.25	0	0	0.25	0.32	2	
Lysiphlebus testaceipes	Hymenoptera	0	0	0.24	0	0	0.25	0.17	2	
Note:

Molecular traits reported include primer mismatch and GC% of the amplified sequence for each of the two primer pairs used. Morphological traits include measurements of insect volume, obtained by averaging measures across 10 specimens per species, as well as a scale of exoskeleton hardness. Hardness values are 1 = soft, 2 = intermediate, 3 = hard.

Molecular analysis

Preparation of insect mock communities

In order to partition total protocol bias into its constituent steps, mock communities of bulk adult insect, genomic DNA, and PCR amplicons were assembled to simulate the input of each major metabarcoding laboratory step.

DNA was non-destructively extracted from the eight pools of insects using both the QuickExtract kit (Biosearch Technologies, Novato, CA, USA), for pools 1–4, and the DNEasy Blood and Tissue kit (Qiagen, Germany; “DNEasy” hereafter), for pools 5–8, following the methods used for single insects in Martoni et al. (2021). These pools, while having a different number of individuals for some of the species, were prepared with an almost identical Order-level composition, taking into account insect biomass and exoskeleton hardness. Non-destructive DNA extraction using QuickExtract was performed as follows: ethanol was removed from the pooled insects and air-dried in tubes for 10 min. Five hundred microlitres of QuickExtract were added to the pooled insects, ensuring all insects were submerged, vortexed for 30 s, incubated at 65 °C for 6 min, vortexed for 15 s and incubated at 98 °C for 2 min. The supernatant containing the extracted DNA was then transferred to a new Eppendorf tube and stored at −20 °C until PCR amplification.

Non-destructive DNA extraction using DNEasy was performed following the first steps of the protocol presented in Martoni, Valenzuela & Blacket (2019) and Bahder et al. (2015). Briefly, ethanol was removed from the insect pools (as above), insects were then submerged in an ATL buffer/Proteinase K mix with a ratio of 9:1 and then incubated for approximately 17 h (overnight) at 56 °C. The supernatant was then removed from the insects (as above), and processed further following the manufacturer instructions (i.e., filter column purification and elution). Finally, pooled insect specimens were resuspended in absolute ethanol and preserved.

PCR amplification was conducted using either fwhF2 (GGDACWGGWTGAACWGTWTAYCCHCC)-fwhR2n (GTRATWGCHCCDGCTARWACWGG) or fwhF2–HexCOIR4 (TATDGTRATDGCHCCNGC), which amplify almost entirely overlapping regions of COI (Vamos, Elbrecht & Leese, 2017; Marquina, Andersson & Ronquist, 2018). The primer pair fwhF2-fwhR2n targets a 205 bp amplicon (excluding primers) from 346 bp to 551 bp within the conventional COI barcode region, while fwhF2-HexCOIR4 target a 214 bp amplicon (excluding primers) from 346 bp to 560 bp. PCR amplification was performed using the Bioline MyFi DNA Polymerase kit (Meridian Bioscience, Cincinnati, OH, USA) using 2.5 µL of DNA template and 1 µL each for the two primers (10 mM) in a 25 µL final volume. The PCR was run with the same cycling conditions for both primer pairs, with an initial 5-min denaturation at 95 °C, followed by 30 cycles of denaturation at 95 °C for 45 s, annealing at 50 °C for 30 s and extension at 72 °C for 30 s, followed by a final extension at 72 °C for 7 min. PCR amplification was verified on a 1% w/v agarose gel.

PCR amplicons were used as template for a second round of PCR to attach Illumina sequencing adapters with unique dual indexes to each sample, using Phusion High-Fidelity DNA Polymerase (New England Biolabs, Ipswich, MA, USA). PCR conditions were an initial denaturation of 30 s at 98 °C followed by 8 cycles of denaturation at 98 °C for 10 s, annealing at 65 °C for 30 s and elongation at 72 °C for 30 s. The adapter tailed and indexed amplicons were purified using AMPure XP beads (Beckman Coulter, Brea, CA, USA) following the manufacturer instructions. Library fragment size (amplicon + adapters) and absence of primer dimers was verified on an Agilent TapeStation 2200 (Agilent Technologies, Santa Clara, CA, USA) and all libraries were equimolarly pooled based on their concentrations as determined by Qubit dsDNA HS Fluorometric Quantification (ThermoFisher Scientific, Waltham, MA, USA). As DNA concentrations in negative controls were too low to be measured, they were pooled at the same volume of the lowest concentration mock community library. The final pooled library was then diluted to 7 pM, spiked with 15% PhiX, and sequenced on the Illumina MiSeq platform using the V2 reagent kit (2 × 250 bp reads) (Illumina, CA, USA).

Preparation of DNA and PCR mock communities

To assemble mock communities representative of the post-DNA extraction stage (Fig. 1; “DNA Pools”), DNA for each of the 16 species was destructively extracted separately from 5–20 homogenised individuals (depending on their size) with the DNEasy kit. The DNA from each insect species was then quantified using a Qubit 2.0 Fluorometer (Thermo Fisher Scientific, Waltham, MA, USA) and pooled together imitating the composition (% relative abundance) of the original insect pools (Fig. 1, “DNA pools”). In order to do this, an arbitrary 4 ng of DNA was used as the unit corresponding to one specimen and PCR products were diluted to a concentration of 4 ng/µL so to have a consistent volume for each pool (400 µL) at a consistent concentration (4 ng/µL) (i.e., 4 ng of DNA for each insect composing the original pools). Libraries were prepared from these pools following the remainder of the metabarcoding protocol used for the whole insect pools.

Figure 1 Workflow of the experiment for the three types of pools.

Insect pools are the result of non-destructive DNA extractions from the pooled insect specimens. DNA pools are the result of DNA that was destructively extracted separately from each insect species then pooled prior to PCR. PCR pools are the result of DNA that was extracted, quantified, and amplified separately from each insect species then pooled before indexing qPCR. All pools were indexed, sequenced and analysed following the same protocol. Each pool was amplified using the two primer sets fwhF2-fwhR2n and fwhF2-HexCOIR4.

To assemble mock communities representative of the post-amplification stage (Fig. 1, “PCR Pools”), DNA extracted from each insect species as above was then amplified separately with each primer pair using the same PCR conditions as the whole insect mock communities. The PCR amplicons were then quantified using Qubit 2.0 Fluorometer (Fig. 2) and diluted to a concentration of 2 ng/µL so to have a consistent volume for each pool (200 µL) at a consistent concentration (2 ng/µL), as outlined above (i.e., 2 ng of DNA for each insect composing the original pools, to reconstitute the original sample relative abundance), and libraries prepared as above. A second MiSeq run (“Run 2”) was performed 3 months after the first one (“Run 1”, which was conducted immediately) containing sequencing libraries generated from the DNA and PCR pools, as well as repeating the original insect pools using the same MiSeq machine and protocol. Run 2 of the insect pools was used to test how well the extracted DNA could be amplified and sequenced following an extended period of storage at −20 °C, a typical temperature often used for long term storage of DNA samples, enabling future analysis on the same samples.

Figure 2 DNA concentration of PCR amplification for each species.

DNA concentration was measured from the PCR amplification obtained for each of the two primer pairs after 30 PCR cycles (A) or 40 PCR cycles (B) on DNA extracted from each single insect species.

Bioinformatics analysis

Raw sequence reads were demultiplexed using bcl2fastq v2.2.0 allowing for no mismatches to the expected index combinations (NCBI SRA acc no: PRJNA767112), then trimmed of PCR primer sequences using BBDuK v38.9 (Bushnell, Rood & Singer, 2017). Sequence quality profiles were used to further filter reads with >1 expected error (Edgar & Flyvbjerg, 2015), or any ambiguous ‘N’ Bases, then remaining sequences were denoised using DADA2 v1.16 (Callahan et al., 2016) with the error model determined separately for each sequencing run. Following denoising, amplicon sequence variants (ASVs) inferred separately from each sequencing run were combined into a single table and any chimeric sequences removed de-novo using the “removeBimeraDenovo” function in DADA2. To further filter any non-specific amplification products and pseudogenes the ASV’s were aligned to a profile hidden Markov model (Eddy, 1998) of the COI barcode region from Piper et al. (2021) using the aphid v1.3.3 R package (Wilkinson, 2019), retaining sequences that met a minimum log-odds alignment score of 100 with a minimum match length of 100 bp. Retained ASVs were then checked for frame shifts and stop codons that commonly indicate pseudogenes (Roe & Sperling, 2007). Taxonomy was determined by aligning ASVs to reference sequences of each taxon used in the mock communities using BLASTn v2.11.0 with a minimum percentage identity of 97% and minimum alignment coverage of 95%. The 52% of ASVs (0.1% of abundance) that couldn’t be accurately mapped were discarded using filtering functions contained in the phyloseq v1.36.0 (McMurdie & Holmes, 2013) and tidyverse v1.3.1 (Wickham et al., 2019) R packages.

Statistical analysis

Differences in the number of ASVs detected between DNA extraction protocols, primer sets, and mock communities representing each workflow step were tested for significance using Analysis of Variance (ANOVA), followed by post-hoc pairwise comparisons with Tukey’s Honest Significant Difference (HSD) test (Tukey, 1977). In order to compare the number of ASVs between each protocol without the confounding effect of differing read depths all samples were rarefied to 100,000 reads before ANOVAs were conducted. Differences in overall quantitative performance between each protocol was measured using the Root Mean Square Error (RMSE) between observed and expected relative abundances, an accuracy metric where smaller values indicate a less biased protocol. Taxonomic bias for each primer and workflow step (Insect pools, DNA pools, PCR pools) was estimated using a linear model of the compositional error (ratio between expected and observed abundances), with the results geometrically centered to be relative to the ‘average’ taxon (McLaren, Willis & Callahan, 2019) and standard errors for each taxon coefficient generated from 1,000 bootstrap resamples. The fit of the bias model for the dataset was evaluated by its ability to predict the observed relative abundances from sequencing using the known relative abundances in the mock communities. The separate bias estimates from each workflow step were then used to partition the total protocol bias (that represented by the Insect pools) into three different components for each primer set (1) DNA extraction, (2) PCR amplification, and (3) sequencing and bioinformatics, as per McLaren, Willis & Callahan, 2019. The change in abundance of each mock community taxon throughout the metabarcoding protocols relative to its starting abundance (Fig. SM1) was then calculated by sequentially multiplying a starting abundance of one by the partitioned bias components for that taxon. Finally, the effects of morphological traits (biomass, sclerotization) and molecular traits (GC% of the whole amplicon, primer-template mismatch) on both the partitioned bias estimates and total protocol bias (represented by the Insect pools) was tested for significance using a second linear model fit to the bootstrapped bias estimates, and the relative influence of each trait on detection efficiency determined from its coefficient in the regression model. All statistical analyses were conducted within R 4.0.2 (R Core Team, 2020) using tidymodels v0.1.3 (Kuhn & Wickham, 2020) packages.

Results

Comparison between non-destructive DNA extraction methods

A total of 7,241,574 and 8,676,552 reads were generated from the first and second MiSeq sequencing runs respectively, consisting of 182 unique ASVs. Each sample received a mean 272,600 (±3,762) reads, ranging from 615,687 reads for the highest sample to 619 reads for the lowest negative control sample, and contained a mean of 26.7 ASVs (±0.119, range: 11–34). When reads that could not be classified to the known mock community members were removed (e.g., chimeras), the mean reads per sample dropped slightly to 263,581 (±4,076; range: 619–614,641) and the mean number of ASVs dropped to 14.9 per sample (±0.03, range: 7–16). Significant differences in the number of inferred ASVs were found between the different protocols (ANOVA; F(3,60) = 5.11, p = 0.003), primarily driven by the DNEasy treatments showing significantly more unique ASVs than the QuickExtract (Tukey’s HSD; p = 0.002). On further exploration this difference was found to be due to a substantial dropout of taxa seen in the replicated QuickExtract samples on Run 2, which was run three months later (Fig. SM1). When these analyses were repeated without the replicated QuickExtract samples, significant differences were again found between the treatments (ANOVA; F(3,52) = 9.78, p < 0.001), driven by both the QuickExtract and DNEasy treatments having higher numbers of ASVs than the DNA pools (Tukey’s HSD; p = 0.006) and the PCR pools (p = 0.002) with no significant difference between the QuickExtract and DNEasy samples (p > 0.05). When considering only ASVs that could be classified to species level, significant differences remained between treatments (ANOVA; F(3,52) = 6.18, p = 0.001), which pairwise comparisons revealed to be driven by the QuickExtract having significantly more ASVs than the DNA (Tukey’s HSD; p = 0.002) or PCR pools (p = 0.001), while no significant differences were found between the DNEasy and any other treatment (p > 0.05). In contrast to the differences in protocols and pools, there were no significant differences between the two primers sets in terms of both total ASVs (ANOVA; F(1,54) = 0.21, p > 0.05) or ASVs that could be classified to species level (ANOVA; F(1,54) = 4.04, p > 0.05).

Identification of the sources of bias

Both QuickExtract and DNEasy showed the largest deviation between expected and observed (from sequencing) relative abundances at 16% and 17% RMSE respectively for both primer pairs (Fig. 3A) when compared to DNA pools (9% for the fwhF2-fwhR2n primer pair, and 7% for the fwhF2-HexCOI4 primer pair) and the PCR pools (7% and 2%). There was a significant association found between the compositional error (ratio between expected and observed abundances) and species identity across all treatments and primer sets (Table S1). When the taxon coefficients estimated by the bias model (Table S2) were used to predict the observed relative abundances from the known number of individuals in the mock communities, a substantial improvement in quantitative performance was seen compared to the uncorrected sequencing data (Fig. 3A). The model showed a good fit to the DNEasy pools, reducing the RMSE to 2% and 3% for the fwhF2-fwhR2n and fwhF2-HexCOI4 primers respectively, and a near perfect fit to the DNA and PCR pools with 1% RMSE for both primers. While the bias model also reduced the RMSE for the QuickExtract samples, there was still substantial variance seen with an RMSE of 9% for fwhF2-fwhR2n, and 7% for fwhF2-HexCOI4 (Fig. SM2).

Figure 3 Comparison of estimated and observed proportion of reads.

(A) Comparison of observed relative abundances to expected relative abundances of each taxon, matching colours shown in part B. (B) Estimated bias for each taxon displayed relative to the geometric mean bias, with 95% and 50% confidence intervals displayed. RMSE, Root Mean Square Error.

Taxon-specific biases

The taxonomic biases estimated by the model (Table S2) revealed A. solanicola, A. alternata, Acizzia sp. and R. padi having the highest efficiency in the QuickExtract and DNEasy pools, and to a lesser extend the DNA pool. On the other hand, the two Carpophilus taxa showed the lowest relative efficiency (Fig. 3B). The bias estimates for the QuickExtract insect pools showed substantially higher variance compared to the DNEasy insect pools across all taxa, with Carpophilus truncatus showing the largest difference between the two extraction methods (Fig. 3B). When the total protocol bias was partitioned into the contribution of each step (Table S3), marked differences in bias were seen between protocol steps on the same taxon (Fig. 4A). For instance, Carpophilus truncatus saw substantially lower DNA extraction efficiency than the average taxon, but higher than average at the PCR stage. In contrast, Diuraphis noxia showed higher DNA extraction efficiency, but lower efficiency in the later PCR step. The results for the PCR amplification are similarly reflected in the DNA concentrations obtained when single species were amplified to obtain mock communities representing post-PCR processes (Fig. 2). Here we see that when individual PCRs are conducted from the same starting DNA concentration, Diuraphis noxia has much lower and Carpophilus truncatus has much higher DNA concentrations than the majority of the other taxa after both 30 and 40 cycles of amplification. The competing effects of different workflow stages on overall detection efficiency is particularly apparent when looking at the change in abundances of molecules for each taxon throughout the workflow (Fig. 4B). Some taxa such as the psyllids A. solanicola and A. alternata consistently increase in abundance throughout the workflow, while others such as the fruit fly B. tryoni saw an initial increase in abundance, followed by a fall to almost equilibrium with the starting relative abundance (Fig. 4B).

Figure 4 Metabarcoding bias partitioned.

(A) Metabarcoding bias partitioned to each protocol and primer step. Change in composition of each taxon throughout the workflow relative to the geometric mean taxon, for (B) QuickExtract, and (C) DNEasy obtained by sequentially multiplying a starting abundance of one by the partitioned bias estimates from panel A.

Hardness- and biomass-associated bias

Differences in whole protocol bias was significantly associated with insect traits for both the QuickExtract (F(5,54) = 5.68, R2adj = 0.26, p < 0.001) and DNEasy (F(5,58) = 20.50, adjusted R2adj = 0.60, p < 0.001) protocols (Fig. 5A). The efficiency with which a species was detected in a sample was positively influenced by whether that species had a soft (QuickExtract; β = 35.53, 95% CI [9.70 to 123.02], DNEasy; β = 15.88, 95% CI [11.86–22.10]), or intermediate hardness exoskeleton (QuickExtract; β = 19.13, 95% CI [6.09–62.83], DNEasy; β = 8.55, 95% CI [6.65 to 11.46]), or high amplicon GC% (QuickExtract; β = 31.93, 95% CI [7.53 to 82.098], DNEasy; β = 13.71, 95% CI [9.90 to 19.57]) (Fig. 5A). Insects with a hard exoskeleton showed no increase in detection efficiency for the DNEasy protocol (β = 0.12, 95% CI [−0.68 to 0.86]), but increased efficiency for the QuickExtract protocol although with a confidence interval overlapping zero (β = 2.29, 95% CI [−0.43 to 8.99]). Insect biomass unexpectedly had a strong negative effect on detection efficiency for both protocols (QuickExtract; β = −12.02, 95% CI [−41.50 to −3.07], DNEasy; β = −4.73, 95% CI [−6.87 to −3.40]), exceeded only by primer mismatch (QuickExtract; β = −23.20, 95% CI [−82.01 to −5.84], DNEasy; β = −8.50, 95% CI [−12.66 to −5.63]) (Fig. 5A). Generally, biases associated with each trait were less predictable (i.e., with larger variances observed) when using QuickExtract (R2adj = 0.26) compared with DNEasy (R2adj = 0.60) extraction method (Fig. 5A). When considering just the bias contributed by the DNA extraction stage (Fig. 5B), insect traits were also associated with detection efficiency albeit to a lesser degree than the whole protocol (QuickExtract; F(5,54) = 8.79, R2adj = 0.38, p < 0.001, DNEasy; F(5,58) = 32.60, R2adj = 0.70, p < 0.001). Most notably, the relationship to insect biomass was reversed when considering just the DNA extraction stage, with larger biomass showing a slight positive effect on detection efficiency for both protocols (QuickExtract; β = 0.94, 95% CI [0.20 to 1.81], DNEasy; β = 2.76, 95% CI [2.15 to 3.51]) (Fig. 5B). A soft (QuickExtract; β = 1.78, 95% CI [1.11 to 2.97], DNEasy; β = 1.85, 95% CI [1.56 to 2.20]) or intermediate hardness exoskeleton (QuickExtract; β = 2.56, 95% CI [1.85 to 3.51], DNEasy; β = 2.79, 95% CI [2.50 to 3.10]) again increased detection efficiency during DNA extraction, while a hard exoskeleton, primer mismatch, or amplicon GC% all had confidence intervals overlapping zero for both protocols (Fig. 5B). When considering only the bias contributed by the PCR stage, the overall effect of insect traits on detection efficiency was more comparable between protocols (QuickExtract F(5,54) = 7.13, R2adj = 0.34, p < 0.001, DNEasy; F(5,58) = 7.65, R2adj = 0.35, p < 0.001) (Fig. 5C). As expected, primer mismatch showed the strongest negative effect on detection efficiency (QuickExtract; β = −3.18, 95% CI [-3.66 to -2.60], DNEasy; β = −3.08, 95% CI [−3.44 to −2.83]), while amplicon GC% increased detection efficiency (QuickExtract; β = 3.15, 95% CI [2.65 to 3.83], DNEasy; β = 3.14, 95% CI [2.86 to 3.50]). All values of exoskeleton hardness showed a slight positive effect on detection efficiency at the PCR stage across both protocols (β 2.37 to 3.09), while biomass decreased efficiency (QuickExtract; β = −0.91, 95% CI [−1.16 to −0.67], DNEasy; β = −0.92, 95% CI [−1.06 to −0.79]).

Figure 5 Model predicting the estimated taxon-specific bias.

Coefficients of model predicting the estimated taxon-specific bias from the traits in panel A for the unpartitioned bias and the three partitioned bias steps. Coefficients are displayed on a pseudo-log scale to avoid compressing around zero.

Discussion

Here, using non-destructive metabarcoding approaches, we successfully recorded all insect species present in pools composed of 100–101 individuals, including those species that were represented by just a single individual insect. At the same time, the use of these non-destructive DNA extraction methods allowed morphological voucher specimens of the insects to be preserved, as previously demonstrated (Martoni, Valenzuela & Blacket, 2019; Batovska et al., 2021). This is of paramount importance for many regulatory applications of metabarcoding, where retaining voucher specimens of potential pests or indicator taxa can be required for legal reasons, or to simply provide a morphological specimen preserved in an entomological collection for future taxonomic investigation (i.e., Martins et al., 2019; Martoni, Valenzuela & Blacket, 2019; Martoni, Valenzuela & Blacket, 2021).

The results presented here show that non-destructive metabarcoding analysis can be successfully applied to bulk samples of agriculturally relevant insects to obtain a species identification, in agreement with recent studies (Carew, Coleman & Hoffmann, 2018; Nielsen et al., 2019; Batovska et al., 2021). This suggests that non-destructive metabarcoding has potential applications not only for biodiversity assessments but also for diagnostics and biosecurity purposes, to determine presence/absence of pests in bulk traps. Our study further highlights the molecular and morphological traits that affect quantification of individual insect species within non-destructively extracted bulk insect pools.

Metabarcoding bias and non-destructive DNA extractions

The output generated by metabarcoding analyses is compositional data (Gloor et al., 2017). This means the relationship between the starting total abundance of a species and the output counts of sequence reads is completely lost, and the sequence reads returned for a taxon are only meaningful relative to the rest of the taxa within its sample (Gloor et al., 2017). Furthermore, the relationship between counts of sequence reads and the individuals they arise from is affected by a number of biases that systematically distort the measured sequence counts of each species from their true abundances (McLaren, Willis & Callahan, 2019). This bias can lead to taxonomic dropouts by diluting certain taxa below the detection limit and brings additional challenges for quantifying the number of individual specimens in a sample from metabarcoding sequences alone.

Amongst the known sources of metabarcoding bias, a range of physical characteristics of the insect community under study plays a key role, with perhaps the most obvious of these being the large variation in specimen body sizes within insect community assemblages (Chown & Gaston, 2010). For example, even assuming that all available DNA is always extracted from a sample, the amount of DNA (and the reads obtained from it) from a single insect will depend on its biomass, which in turn depends on its species or life stage. Consequently, when aiming to use metabarcoding as a semi-quantitative technique, it is important to remember that the reads obtained from a single insect will vary depending on a number of its characteristics (Thomas et al., 2016; McLaren, Willis & Callahan, 2019), as well as on the composition of the insect pool analysed (Gloor et al., 2017).

With non-destructive DNA extractions, an additional factor to consider is that non-destructive protocols mostly act upon the external surface of the insect exoskeletons, as opposed to destructive methods that can potentially access all of the DNA contained in the insect bodies. Therefore, when comparing the number of reads obtained from the insect pools with that from the DNA pools two additional factors had to be taken into consideration in addition to sample composition: exoskeleton robustness and species biomass. When comparing these factors, a soft exoskeleton had the strongest positive effect on relative efficiency, suggesting that soft-bodied insects facilitate non-destructive DNA extraction, resulting in a higher representation than one would expect from their relative biomass within the sample. Whether this proves beneficial or problematic for species detection will depend on the specific communities a surveillance programme is targeting. On the one hand, the lesser influence of species biomass could alleviate the requirement for any morphological pre-sorting to ensure large specimens don’t drown out smaller ones (Elbrecht, Peinert & Leese, 2017). However, the negative effect of hard exoskeletons on detection efficiency could produce false negatives for low abundance taxa with a high level of sclerotization. This is the most likely explanation for the differences in extraction efficiency between taxa such as Carpophilus truncatus, having a highly sclerotized exoskeleton, in comparison to the more soft-bodied insects such as Diuraphis noxia (Fig. 5A).

These results were consistent between both the QuickExtract and DNEasy kit protocols, suggesting that the bias towards softer bodied insects is an inherent aspect of the non-destructive DNA extraction process itself, rather than a specific protocol or kit. Nevertheless, there are a few key differences between the two extraction methods we evaluated that will influence their practical application within future studies. Firstly, the QuickExtract method was substantially faster, requiring only 1 h of operator’s time as opposed to the overnight incubation of the DNEasy method. However, QuickExtract produced unpredictable variation in DNA extraction performance and appeared less suitable for long-term DNA preservation. When the DNA extracts from the insect mock communities were re-sequenced three months after extraction–during which the templates were kept in −20 °C freezer–the QuickExtract template appeared to have degraded dramatically, resulting in a number of species dropping out. On the other hand, the presence/absence of species from the DNEasy kit was virtually identical to when they were sequenced within days of being extracted. Therefore, DNA extraction products obtained with the QuickExtract kit should be used immediately for analysis, or stored at lower temperatures, i.e., in a −80 °C freezer (as suggested in the manufacturer’s recommendations) if possible.

In addition, the results showed relative abundance measurements obtained through non-destructive metabarcoding assays are most strongly influenced by the DNA extraction processes. DNA extraction was not just the largest contributor to protocol bias, but also had the highest variance of all the tested workflow stages, making it less predictable. Nevertheless, it is unclear if this aspect is specific to the non-destructive DNA extraction process or is applicable to DNA extractions generally. Few metabarcoding studies have attempted to partition the total protocol bias into separate components for each major protocol step. It would be valuable for future research to compare the quantitative results of destructive and non-destructive extractions, using a similar bias-partitioning approach to that implemented within our study. The non-destructive DNA extraction bias issues highlighted here may turn out to be unavoidable when voucher specimens are required, as destructive DNA extraction is not an option.

Primer mismatch and PCR bias

Beside DNA extraction, we wanted to assess if bias was introduced also at the PCR stage. Mismatches between PCR primers and template molecules have been considered the primary contributor to metabarcoding bias, particularly at the 3′ end of the primer where nucleic acid extension takes place (Piñol et al., 2015). Primer-template mismatch can be particularly problematic for protein coding genes such as COI, due to natural degeneracy in the genetic code leaving no strictly conserved gene regions for design of universal PCR primers (Deagle et al., 2014). This has necessitated the inclusion of multiple degenerate nucleotide bases in metabarcoding primers in order to mitigate the effects of mismatch on detection efficiency and abundance estimates (Elbrecht et al., 2019).

While much of the previous literature has highlighted that bias introduced during PCR amplification is responsible for the semi-quantitative nature of metabarcoding results, our study observed a smaller contribution of PCR compared to the DNA extraction process. While this could potentially be attributed to non-destructive DNA extraction being more bias prone than its homogenisation-based alternatives, this will require further experiments to demonstrate. Alternatively, the smaller contribution of the PCR process to total protocol bias could have been due to the two primer pairs employed in this study containing a number of degenerate bases, having been designed to be generic for insects (Vamos, Elbrecht & Leese, 2017; Marquina, Andersson & Ronquist, 2018; Elbrecht et al., 2019). Previous studies that have found high levels of PCR bias have mostly used less-degenerate primers or primers which have since been identified to contain critical design flaws (Elbrecht & Leese, 2017; Piper et al., 2021). Nevertheless, some species included in our study still showed mismatch to these generic primers (Table 2), and this was a primary driver of reduced efficiency at the PCR stage (Fig. 5C).

Additionally, primer bias was confirmed when quantifying the PCR product obtained from each insect species in separate PCRs, where each primer pair amplified the DNA of some species up to 10 times more than others. Since this DNA had been normalized prior to PCR, variation in subsequent concentration could be linked to bias introduced during amplification. Such variation in the concentration appeared to also depend on the number of PCR cycles performed, with a higher number of cycles showing more similar concentrations across different insect species. Nevertheless, this bias did not impact the presence/absence of targets, with both primer pairs tested here recording all of the species present in all the pools, with a sensitivity of up to 1 in 101 for many of the species tested. Further work could be required to test if primer bias may have greater effects on presence/absence in larger communities, or at a lower sequencing depth, as well as if alternative primer pairs might produce a lower primer bias. Nonetheless, it is important to remember that, in a diagnostic context, priority is given to a precise presence/absence assessment, especially when testing for the presence of unexpected pests or rare species. Therefore, for general insect biodiversity monitoring and surveillance, the semi-quantitative but taxonomically broader screening of agricultural traps or biodiversity assessment samples that can be performed using MiSeq metabarcoding, could be considered the preferred approach over a more quantitative but perhaps more time-consuming method, such as qPCR, targeting each individual species separately.

Possible strategies to account for metabarcoding bias

When considering the number of reads generated as output of the MiSeq metabarcoding, the relative abundance of reads obtained for each insect species was correlated with the number of individuals present in the pool but biased toward certain species. Following the multiplicative model of McLaren, Willis & Callahan (2019), we used mock communities representing each step of the library preparation pipeline to partition the total protocol bias into that explained by each component. Indeed, the relative abundances obtained from different insect species appeared to be subject to bias during the different steps of the laboratory workflow, including DNA extraction and PCR, fundamentally due to both morphological and molecular traits differing across each insect group. Amongst these, some of the morphological traits determined here to significantly affect the metabarcoding results are surface area, size and consistency of the insect tissues, as previously reported for Coleoptera (Martins et al., 2019). On the other hand, molecular traits include PCR primers mismatch, which was observed here for both the primer pairs used, despite these being considered generic primers (Vamos, Elbrecht & Leese, 2017; Marquina, Andersson & Ronquist, 2018). In the literature, attempts have been made to improve a quantitative output for metabarcoding, either by developing new primers, or protocols (e.g., Elbrecht & Leese, 2015; Deagle et al., 2014; Lamb et al., 2019). Our results indicate that understanding the individual contributions of each laboratory stage, rather than just the total overall protocol bias, should be a critical consideration for future efforts. This was particularly apparent when measuring the effect of biomass on detection efficiency, as when the total protocol bias was considered, larger biomass appeared to have a strong negative effect on the number of reads for each given species. On the other hand, when considering just the bias introduced by the DNA extraction stage, larger insect biomass increased the number of reads produced, as expected from previous studies. Therefore, when optimizing protocols researchers should be wary of the often-contrasting effects of the different laboratory steps, in order to not be confounded into mistakenly attempting to optimize one aspect, when in reality the majority of the bias is being introduced through a different laboratory step.

While efforts to further optimise protocols will no doubt prove important for increasing the quantitative performance of metabarcoding, a less explored but complementary approach is to use statistical models to actively correct for bias during analysis (Thomas et al., 2016; Krehenwinkel et al., 2017; McLaren, Willis & Callahan, 2019). Promisingly, we found bias to be highly predictable with our model, demonstrating the potential for developing correction factors to improve the quantitative results of metabarcoding assay for specific important target species, e.g., agricultural pests. However, even with our consistently treated mock communities, DNA extraction bias had the highest variance, and thus would be the most difficult factor to model and correct for. In a surveillance situation, this could be exacerbated by differences in environmental conditions between real trap samples and the mock communities used to derive the correction factors introducing further bias (Krehenwinkel et al., 2018). Furthermore, developing correction factors from mock communities requires knowing the organisms that will likely be encountered a priori, as well as being able to acquire specimens of them. Unfortunately, in both contexts of biodiversity monitoring and surveillance, knowing a priori what organisms will be recorded in order to prepare targeted spike-ins or tailored mocks community could be challenging and, often, defy the very purpose of the study.

Conclusions

Based on the results obtained here, different insect species, even belonging to the same order or genus, are subject to different degrees of bias during a non-destructive metabarcoding workflow. These biases are driven by species-specific morphological and molecular traits that interact with different protocol steps to either increase or decrease detection efficiency. If bias can be measured for that taxon a priori using mock communities, correction factors could therefore be used to calibrate the results to better reflect the actual abundances, but requires knowledge of the sample composition prior to analysis.

Ultimately, if the surveillance program or the biodiversity assessment are conducted long-term using a non-destructive DNA extraction and thus retaining voucher specimens, obtaining specimens to develop correction factors covering a wider diversity may eventually be achievable. Inclusion of mock communities in the metabarcoding analysis would not only be possible, but would sensibly improve in accuracy based on an ongoing monitoring of the geographical area of interest. This highlights the importance of long-term monitoring and biodiversity data collection projects not only for agricultural and industrial areas but also for adjacent ecosystems that can contribute to the species diversity recorded during insect trapping programs.

Supplemental Information

Supplemental Information 1 Comparison of non-destructive DNA extractions.

Comparison of quantitative (A) and qualitative (B) results for DNA extraction methods on whole insect pools in Run 1, sequenced immediately, compared with Run 2, after 3-months storage at −20 °C.

Click here for additional data file.

Supplemental Information 2 Fit of the bias model to each set of pools.

Proportions are displayed on a pseudo-log scale to avoid compressing variation around zero. RMSE; Root Mean Square Error.

Click here for additional data file.

Supplemental Information 3 Overall model fits across the primers and mock community types.

Click here for additional data file.

Supplemental Information 4 Estimated coefficients (bias) for each taxon, with bootstrap standard errors across the primers and mock community types.

Click here for additional data file.

Supplemental Information 5 Partitioned bias, and bootstrap standard errors across the primers and mock community types.

Click here for additional data file.

The authors thank PeerJ editor, Dr Sheila Colla, and two anonymous reviewers for their useful comments and suggestions. We thank Isabel Valenzuela, Lea Rako, Jessi Henneken, and Will Boston (Agriculture Victoria) for providing many of the laboratory colony insect specimens used in this study.

Additional Information and Declarations

Competing Interests

Author Contributions

Data Availability

The authors declare that they have no competing interests.

Francesco Martoni conceived and designed the experiments, performed the experiments, analyzed the data, prepared figures and/or tables, authored or reviewed drafts of the paper, and approved the final draft.

Alexander M. Piper conceived and designed the experiments, analyzed the data, prepared figures and/or tables, authored or reviewed drafts of the paper, and approved the final draft.

Brendan C. Rodoni conceived and designed the experiments, authored or reviewed drafts of the paper, and approved the final draft.

Mark J. Blacket conceived and designed the experiments, authored or reviewed drafts of the paper, and approved the final draft.

The following information was supplied regarding data availability:

The raw data is available at NCBI SRA: PRJNA767112. The code used is available at GitHub: https://github.com/alexpiper/disentangling_metabarcoding_bias_MS.

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
