# Peer review of "Disentangling bias for non-destructive insect metabarcoding"

_PeerJ, doi:10.7717/peerj.12981_

## Round 0.1 · original submission · Minor Revisions

Thank you for your submission. I apologize for the delay. I am pleased to recommend it for minor revisions and look forward to the resubmission.

Reviewer 1 ·

Basic reporting

The manuscript is well written, although there are few minor edits to be made. The tables and figures keep changing the order that the taxa are presented, which puts undue strain on the reader to keep trach of the different species used. The figure legends are often too brief without providing sufficient detail. At times I wondered about the article structure. It's okay, but not great. There are a few different aspects of metabarcoding being examined here, but I was constantly trying to establish what was being tested, shown at each step. It might be useful to add more structure to the text where you explicitly state each aspect and deal with them in turn. DNA extraction method. Primers. Insect/DNA/PCR pools. It's all in there, but I had to step away from the article and when I came back I needed to back through all the methods to figure out was going on. Could be structured to be more intuitive following the Figure 1 structure. seems like most of the code is on github and the DNA will be posted, but I didn't see the latter available.

Experimental design

An interesting design. The molecular methods and bioinformatics seem appropriate. The root mean square error is fine, but some context on what a desirable result for this may be useful. It kind of depends on the application I think. Some details in the molecular methods may have been left out because they are so common, but please reference any standard protocols that you use and be explicit regarding any deviations. The layout of the known pools to test how biases at different steps effect results is useful. The degree of sclerotization is probably about right, but seems a bit hand wavy. / subjective. The volume measurements are also unclear. They are taken from a stacked (2D) image, but it's impossible to know how this was extrapolated into volume. Were multiple images taken so that L x W x D could be taken? what body part was used for these? The statistical model doesn't provide some of the usual tests I expect to see regarding model assumptions, independent variables, model fit, etc. This could be improved.

Validity of the findings

I had a couple takeaways. 1) non-destructive DNA extractions can work for metabarcoding (good stuff!)
2) metabarcoding has persistent and difficult to predict biases that keep it from being useful for measuring abundance, although presence/absence is effective.

These are both useful for areas of research where metabarcoding may be considered. I'm not particularly encouraged that its a useful technique for areas where abundance does matter, which is unfortunate.

The findings seem valid based on the data / stats presented (although if this is not he main conclusions the author wants to make then maybe some more clarity is in order).

Additional comments

I made a few minor comments on attached version of the ms.

Annotated reviews are not available for download in order to protect the identity of reviewers who chose to remain anonymous.

Reviewer 2 ·

Basic reporting

Intro and background provide good context to the study, literature is well-referenced and relevant and the English language used throughout is clear and professional.

Figures are relevant however the quality is low (increase resolution/size of points on figures 2,3, and 4), however this may be the saving to PDF process

SRA supplied for raw data but no number given so could not access raw data

Experimental design

This study is within the scope of the journal and includes rigorous investigation performed to a high technical standard, methods are well-described, and the research question is well-defined, relevant and meaningful.

Validity of the findings

Novel study with benefit to literature clearly stated. Data SRA provided but no number so cannot access.

Strong conclusion and links to original research question in addition to suggesting future research ideas

Additional comments

Fantastic study – I very much enjoyed reviewing this manuscript and feel it will be a great addition to the literature.

---

## Round 0.2 · accepted · Accept

Many thanks for addressing all the comments provided by the reviewers. I am happy to let you know the manuscript is now accepted for publication.